# Living the ‘Best Life’ or ‘One Size Fits All’—Stakeholder Perceptions of Racehorse Welfare

**DOI:** 10.3390/ani9040134

**Published:** 2019-03-31

**Authors:** Deborah Butler, Mathilde Valenchon, Rachel Annan, Helen R. Whay, Siobhan Mullan

**Affiliations:** School of Veterinary Sciences, University of Bristol, Langford, Somerset BS40 5DU, UK; mathilde.valenchon@bristol.ac.uk (M.V.); rachel.annan@bristol.ac.uk (R.A.); Bec.Whay@bristol.ac.uk (H.R.W.); Siobhan.Mullan@bristol.ac.uk (S.M.)

**Keywords:** racehorse welfare, health, horse-human relationship, focus groups, thematic analysis

## Abstract

**Simple Summary:**

British horseracing industry stakeholders were asked to discuss their perceptions of racehorse welfare. From the discussions held with stakeholders eight different areas that would have an effect on welfare were pinpointed, with health as the most important. Two strands ran through all eight identified areas. These were health-related factors and the horse-human relationship. In their view, to live the ‘best life’ possible a horse in training should be treated as an individual whereas a ‘one size fits all’ approach best fitted a life lived where minimum welfare standards were in place. Participants highlighted some of the challenges racehorses face in terms of welfare together with any innovative or uncommon practices they had seen used. Health was seen as the most important challenge to welfare as well as being the most innovative, for instance, the continued growth in specialised veterinary treatments. The results from this study can be combined with practical animal welfare evidence to produce the first British racehorse welfare assessment tool.

**Abstract:**

The purpose of the study was to explore the perceptions held by British racing industry stakeholders of factors influencing racehorse welfare. Ten focus groups were held across the UK with a total of 42 stakeholders from a range of roles within racehorse care including trainers, stable staff and veterinarians. Participants took part in three exercises. Firstly, to describe the scenarios of a ‘best life’ and the minimum welfare standards a horse in training could be living under. Secondly, to identify the main challenges for racehorse welfare and thirdly, to recall any innovative or uncommon practices to improve welfare they had witnessed. Using thematic analysis, eight themes emerged from the first exercise. Two strands, factors that contribute to maintaining health and the horse-human relationship ran through all eight themes. Across all themes horses living the ‘best life’ were perceived as being treated as individuals rather than being part of a ‘one size fits all’ life when kept under minimum welfare standards. Health was both perceived as the main challenge to welfare as well as one open to innovative practices such as improved veterinary treatments. Data obtained, informed by the knowledge and expertise of experienced stakeholders, combined with practical animal welfare science will be used to develop the first British racehorse welfare assessment protocol.

## 1. Introduction

British thoroughbred racing and breeding is said to generate £3.45 billion for the economy and is the country’s largest sport behind football in terms of attendances, employment and revenue generated [1]. It is overseen by the British Horseracing Authority (BHA) whose responsibilities include the governance, administration and regulation of horseracing. One of their key roles and one of the ways the racing industry differs from the wider equine industry is that stakeholders who are directly involved with the racing industry must be registered, for instance, in the case of owners and stable staff and licensed, for instance, in the case of trainers and jockeys. This means the BHA have the power to ban individuals who seriously breach the rules and regulations pertaining to the racing industry, such as race fixing. 

The racing industry is inextricably linked to two other organisations, British thoroughbred breeding and the betting industry. The breeding industry supply a large proportion of the bloodstock for racing. Owners, racecourses and the betting industry (bookmakers) supply the funding. Bookmakers are required to pay a percentage of their gross profits from their horserace betting business to the Horserace Betting and Levy Board in the form of a statutory levy, 90% of which is then used for the improvement of horseracing [2]. 

In 2018 there were just under 600 licensed racehorse trainers training 23,599 racehorses per annum, who ran on average 5.71 times on the flat, 4.22 times in National Hunt races [3]. Racehorse training yards are located over the length and breadth of Britain and vary in size from less than 10 horses in training to over 250 [4]. Each trainer will have their own methods of getting horses fit which, to some degree, will be dependent on the training facilities they have. Feeding methods and the types of concentrates and forage fed will also vary but usually horses will be fed concentrates three to four times a day with access to forage for part or all of the day. All racehorses must have access to an individual stable of minimum 13.3 m^2^ [5] and many will be housed in these stables when not training or racing. The use of turnout for racehorses, either individually or in groups, is used by some trainers, although it is not a universal practice due to a perception of a risk of injury and limited access to suitable turnout. Horses are transported to race meetings by lorry then stabled in racecourse stabling. 

To date nothing is available for assessing the welfare of racehorses in training in Great Britain although several protocols have been developed to assess horse welfare [6,7,8]. The range of frameworks and the tools currently available both at individual and population level for assessing the welfare of the wider equine population as described in research literature have been reviewed by Hockenhull and Whay [9]. As part of the AWIN (Animal Welfare Indicators) project Dalla Costa et al. [10] have assessed the scientific literature to find potential animal- based welfare indicators including a Horse Grimace Scale and Qualitative Behaviour Assessment, yet very few of these frameworks are racehorse specific. Mactaggart, [11] developed a Thoroughbred Racehorse Welfare Index for racehorses in Australia, as yet it is not possible to know how this was received by the racing industry there. 

To be able to assess and improve racehorse welfare there is currently a need for a holistic approach to welfare assessment that incorporates expert opinion and scientific evidence [12,13]. It is for this reason that racing industry stakeholders were consulted, using focus groups as a method to discover their insights, experiences and their perceptions of factors important in maintaining and improving the welfare of racehorses in training (see Table 1 for occupational backgrounds of stakeholders). A focus group involves engaging a small number of people in an informal group discussion (or discussions), around a specific topic or set of issues [14]. Participants are able to discuss perceptions, opinions, ideas and also to disagree with others over particular points.

Lazarfeld and Merton were credited with formalising the method used for focus groups [15] as a way to identity ‘salient dimensions of complex social stimuli as [a]precursor to further quantitative tests’ [16]. Focus groups have been used across a wide range of topics: with consumers [17]; Lassen et al. [18] consulted with lay participants on what participants perceived as ‘good’ pig welfare. Miele et al. [19] used focus groups as a method to incorporate inputs from both society and science when developing farm animal welfare assessment tools which could be used to assessing animal welfare. Focus groups were used in a similar way by Horseman et al. [20] and Collins et al. [21] when exploring stakeholder’s perceptions of equine welfare assessment and solutions to equine welfare problems in Ireland respectively. Ascertaining the views of experienced stakeholders in all equine fields in this way begins to open up discussion of often neglected areas. The aim of this study was to explore perceptions of racing industry stakeholders with regard to racehorse welfare so enabling the co-creation of a racehorse welfare assessment tool.

## 2. Material and Methods

### 2.1. Participant Recruitment and Response

Participants with experience in both National Hunt (jump) and Flat racing were recruited to reflect the main stakeholder groups within the racing industry (see Table 1). In total, 10 focus groups were held in which 42 participants took part, 22 female, 20 male, representative of the gender balance within stable staff and the veterinary industry.

Potential participants were contacted through a variety of methods. One of the authors (D.B.) worked in racing for many years and was able to draw upon personal contacts for stable staff and ancillary racing stakeholder groups, who in turn asked their friends if they would like to attend. Such an approach could best be described as a snowballing technique [22]. Other participants were contacted through their place of work via email. The focus groups were posted on Facebook by Racing Welfare, the racing charity who support the workforce of British horseracing. Posters were also put up by Racing Welfare in Newmarket and Middleham. As an incentive food and drinks were provided. 

The three main racehorse training centres in Great Britain having communal gallops: (Newmarket, Middleham and Lambourn) were targeted. Holding groups in the main racehorse training areas and centrally in Britain ensured that theoretically, attitudes and perceptions of welfare could be gathered from a range of racing industry participants. It was important to include all key stakeholders within the racing industry which were identified as trainers, stable staff, owners, veterinary surgeons, paraprofessionals such as farriers or physiotherapists, BHA inspectors and human and animal-centred non-governmental organisations. To facilitate a more open discussion two separate focus groups were held in each training centre, one for trainers and one for stable staff and ancillary racing personnel who had worked in racing. These were held in either local pubs or hotels. In addition, a combined focus group was held in the Midlands as there are a plethora of racing yards in these areas, one in London for representatives from equine charities and BHA veterinary officers, another at a University Veterinary School for members of their Equine Veterinary Hospital who work with racehorse trainers and as specialists in the Equine Hospital and the tenth was in the Cotswolds for BHA stable inspectors, BHA veterinary officers who work at the racecourse and paraprofessionals who work with racehorses. Racing’s stakeholders are located across the length and breadth of Britain; focus groups were thus a more resource-efficient way of reaching a wider spread of participants when compared to carrying out face to face interviews. 

### 2.2. Structure of Focus Group Discussion.

Participants were given a participant information sheet and signed a consent form which informed them of their right to withdraw from the focus group. The study and consent process had been given ethical approval by the University of Bristol Faculty of Health Sciences Ethics Committee. 

Each focus group was run by at least two members of the research team, including a trained facilitator (S.M.) for nine of them.

Participants worked together on three exercises. The first scenario-based exercise asked participants to imagine themselves as a racehorse in training. As a racehorse, participants were asked to identify within their group what important elements would form the minimum welfare standards they might be kept under, and conversely what would contribute to the ‘best life’ they, as imaginary racehorses might experience, where money and other factors were no object. These were written down on sticky notes by participants as they discussed the scenarios together and placed on a flip chart paper divided into two sections marked ‘minimum welfare standards’ (MWS) and ‘best life’ (BL).

The second exercise asked participants to individually identify and rank the three main welfare challenges racehorses faced. The challenges were written on sticky notes and put up onto a large flip chart to facilitate a group discussion. 

The final exercise invited participants to individually identify any innovative or uncommon practices that they thought were beneficial to racehorse welfare. Again, these were written on sticky notes and then discussed as a group. 

### 2.3. Data Analysis

Focus groups were audio recorded and transcribed. Focus group data in the form of sticky note statements relating to minimum welfare standards and ‘best life’ were analysed using thematic analysis [23]. Thematic analysis is a qualitative research method for identifying, analysing, organising, describing and reporting themes or categories found within a data set [24]. Once the statements had been tabulated and counted by one of the authors (D.B.) three independent members of the research team (D.B., M.V., R.A.) sifted through them placing them into themes which were then reassembled after discussion to produce the final set of themes (see Table 2). The first author divided the statements allocated to each theme into sub-themes. For example, in the theme health sub categories of collaborative individual health care, veterinary care and farriery were identified within the ‘best life’ scenario (see Appendix A for statements produced for exercise 1).

The statements participants had highlighted as challenges were also analysed using thematic analysis and allocated to the final set of categories. These were then ranked according to the number of statements per theme (see Appendix A for perceived challenges).

Any innovations or uncommon practices that were suggested have been thematically analysed and ranked according to which theme they fitted into.

## 3. Results

The results shown below draw upon an analysis of the 339 statements received in total for all the exercises participants took part in. There were 137 ‘best life’ statements, 99 ‘minimum welfare standards’ statements, eight as standard procedures, 68 as challenges and 27 statements related to innovations and uncommon practices.

For exercise one, participants produced 244 statements in total based on their perceptions of what they thought were minimum welfare standards (99 statements, 41%), the ‘best life’ a horse in training can live, (137 statements, 56%) and standard procedures (8 statements, 3%). Thematic analysis of the statements identified eight key themes (see Table 2) with the highest number of statements (58, 24%) relating to the theme of health. There were more ‘best life’ statements in six of the eight themes than minimum welfare standards. Eight statements relating to policy and procedures were allocated to standard procedures. There were no minimum welfare standards. (See Appendix A, Appendix A for statements produced from exercise one) 

For exercise two, of the 68 statements identified as challenges, health again ranked 1st (18 statements). (See Appendix A, Appendix A for statements produced for exercise two).

One focus group only identified one challenge which related to the value of racehorses after their career had finished. 

In exercise three, two of the focus groups could not identify any uncommon or innovative practices. This was an area many of the groups struggled with, suggesting areas that had been identified as ‘best life’ practices such as the use of additional therapies. There were 27 areas identified in total, these have been analysed and allocated to one of the eight themes (see Table 3). The highest number of statements again (13) related to health. There were no statements within this exercise that corresponded to daily routine and monitoring, feeding, staff management and education and policy and procedures.

The eight themes have been ranked according to the number of statements assigned to each theme.

One of the reasons for using focus groups for this study was to draw upon stakeholder’s experiences of working with racehorses in training. Whilst there were some slight differences of opinions there was, in general, an overall consensus as to what would constitute ‘minimum welfare standards’ and the ‘best life’ a racehorse in training could live. 

One of the key differences was the focus on the individual horse in the ‘best life’ scenario. Minimum welfare standards were perceived and described as a general, ‘one size fits all’ approach. We have used some of the statements from each theme to illustrate this finding within the results.

### 3.1. Health

#### 3.1.1. ‘Best Life’ Scenario

Veterinary care was strongly associated with health care, with ‘... a collaborative preventative healthcare [plan] for individual horses’ with treatment ‘evidence-based not anecdotal’. 

Procedures such as being ‘gastroscoped twice yearly for ulcers’ were seen as part of a healthcare plan with ‘constant veterinary care’ available.

Whilst horses were to be treated as ‘individuals in terms of health care’ participants highlighted the importance of ‘teamwork (farrier, physio, dentist etc)’ with ‘a farrier who knows the horses’ feet’. Part of the emphasis put on teamwork was substantiated through using para-professionals, such as ‘regular physio and chiro (massage etc)’.

#### 3.1.2. Minimum Welfare Standards Scenario

In a minimum welfare standards scenario participants thought that ‘the welfare of horses [was] paramount, vet, essential for injury and disease’. However, the main difference between scenarios was veterinary care, associated with the use of a ‘vet in emergency, vet care when needed’, ‘vet care basic and only when needed’, the converse to veterinary care in ‘best life’. Health care and farriery were perceived similarly. Healthcare was perceived as a ‘general health check’ with a ‘farrier 6 wks or when lost a shoe’, and ‘shoeing before a race and when needed’.

Performance-related veterinary intervention such as ‘the unnecessary use of wind operations’ [surgical procedures to improve upper airway obstructions in horses] and the ‘overuse of medication to keep a horse racing’ seen as indicative of this scenario.

### 3.2. Training, Exercise and Recovery

#### 3.2.1. ‘Best Life’ Scenario

In this scenario some form of exercise would be daily where work routines could be varied, an ‘exercise regime to suit individual horse’ with ‘some form of “cross training” (conditioning exercise not specific to racehorse training) [for example basic flat work (elementary schooling exercise) and ‘varying the working routine for the horses’ mental and physical wellbeing’]. When horses were out training, that is getting fit by galloping, participants emphasised the need for ‘high-quality gallop surfaces to reduce injuries’, with ‘no queuing to get onto the gallops’ when communal gallops were used. 

Horse-human education was perceived as an important factor where ‘riders were to be assigned to ride horses where they suited the horses individually, in terms of rider temperament’. In other words, riders had to be matched to the horse dependent on the horse’s personality rather than the horse matching the rider.

Raceday experience included travelling horses to the races which should be in a ‘quiet, temp [erature] controlled [vehicle], water/hay, adequate travelling staff. One person familiar with the horse’. ‘Climate control in horseboxes’ was suggested as an innovative practice.

#### 3.2.2. Minimum Welfare Standards Scenario

Horses need to be turned out ‘being exercised or ridden six days’ although there was no differentiation in the type of exercise which may also include the use of the ‘[horse] walker every day, alternate direction every other day’. 

Horse-human education was an area which featured more in minimum welfare standards rather than as part of a ‘best life’. That the need for practices, for example, the ‘education of horses re stalls, education at home’, were made explicit may be explained by the fact that they are fundamental lessons for a racehorse and so should be done as a minimum. 

### 3.3. Physical Comfort/Living Environment

#### 3.3.1. ‘Best Life’ Scenario

Emphasis was put on creating a lifestyle tailored to the individual horse with regard to accommodation, with ‘housing suited to individual’s character and some in barns, some set apart, some closed in, some opened up’ with ‘a deep straw bed and mucked out twice a day’ and ‘well ventilated stables’.

Rugs were to be ‘used appropriately depending on the weather conditions’ with each horse having ‘dedicated tack, with regular saddle checks’. The use of ‘tack fitted and tailored to each horse’ continued the emphasis on treating each horse as an individual that is common through all the statements within a ‘best life’ scenario. 

#### 3.3.2. Minimum Welfare Standards Scenario

A minimum welfare standards scenario was more generalised, ‘minimal basic bedding’, ’limited shavings, (where staff are limited on the amount of bedding they can put in), safe, clean, dry, gd light, size enough to roll’. Stabling should be big enough for a horse ‘to get up and down’, but be a ‘safe stable, enough bed so can’t make contact with the floor’ with ‘adequate daylight and ventilation’ and ‘airflow’ with a ‘rug for each weather condition’ and ‘adequate fitted tack’.

### 3.4. Feeding

#### 3.4.1. ‘Best Life’ Scenario

The main distinction between the two scenarios was an economic one when feeding was discussed. Concentrates and forage were perceived as being ‘top quality feedstuffs and supplements’, with ‘top quality forage sourced from anywhere in the world’. 

Emphasis was put on feeding the individual where a horse was given ‘*ad-libitum* forage (tailored to individual)’ and fed ‘specialist feed, (tailored to each horse)’ that was fed as a ‘minimum 3x daily regular balanced diet and appropriate formula for work done’. Water was seen as a necessity either as ‘*ad-lib* clean water’ or as ‘automatic water’ as in automatic drinking bowls.

#### 3.4.2. Minimum Welfare Standards Scenario

Feeding methods were more generalised although it was suggested that minimum welfare standards were the same for ‘best life’, ‘...liberal roughage, *ad-lib* hay, hard feed based on current exercise’. ‘Safe food, not out of date’, ‘concentrates and roughage’, ‘sufficient feed and haylage’ and ‘good quality clean forage and feed (not harmful)’. The times horses were fed was perceived as less flexible with, ‘no variety in (rigid) routine of feeding’, as well as ‘severely restricted forage and too much conc[entrates] proportionately’, indicative or standards below the minimum. ‘*Ad-lib* clean water’ was viewed as a necessity although no one suggested any other method for providing water.

### 3.5. Daily Routine and Monitoring

#### 3.5.1. ‘Best Life’ Scenario

Routine was highlighted as an important part of a ‘best life’ and minimum welfare standards scenario. The difference between the two was the emphasis that for the horse living the ‘best life’ they should be treated as individuals where routines can be changed ‘... if it’s not working’, a ‘great ridden routine, turn out, swimming (tailored to each horse)’.

The horse-human relationship in a ‘best life’ scenario concentrates on individual attention with the ‘same person looking after and riding the same horse every day if they get on’ reflected similarly as ‘consistency of routine and carer’. The ratio of staff to horses was seen as a key part of a ‘best life’ as indicated by the desire to have one member of staff looking after three horses providing a greater opportunity for ‘attention to detail when looking after the horse daily’.

#### 3.5.2. Minimum Welfare Standards Scenario

Minimum welfare standards representative of routine and monitoring were where yards are ‘treating all horses in the same way so one size fits all’ where there are ‘minimal staff and time spent with horse’. The staff: horse ratio suggested of ‘one member of staff to six horses’ was perceived as a minimum as horses ‘don’t get routine care at morning and night, everyone too busy’ although having a ‘routine’ was perceived as important.

### 3.6. Policy and Procedures

#### 3.6.1. ‘Best Life’ Scenario

‘Policy’ as a sub-theme has been used to denote suggestions participants thought could be introduced within yards and by the BHA. In terms of wider racing industry policy it was proposed there should be, ‘an agreed set of interventions for racing clearance post stand-downs’. What was implied was that when a horse has fallen or pulled up they should be prevented from running again until they had been checked by a racecourse vet using an agreed protocol and ‘some assessment of horses at the races that covered pre and post-race management which included warming up and cooling down’.

At present in order to be eligible for a trainer’s licence potential trainers must attend the three mandatory modules based around different aspects of training racehorses. It was thought there should be a change in policy regarding the education of trainers, making the trainer’s modules made more robust, through, for instance, ‘CPD [continuous professional development] for trainers, would have a measurable impact on welfare’. 

In terms of policy relating to racehorses the ‘need for a framework that would monitor and assess racehorse welfare’ was emphasised. 

#### 3.6.2. Minimum Welfare Standards Scenario

There were no minimum standards suggested.

#### 3.6.3. Standard Procedures

Suggestions included procedures based within training yards and were linked to health care. These included having some type of monitoring system within a yard which would record health, daily vet care and level of work, for instance, ‘structured levels of preparation with measurable levels that can be quantified’.

### 3.7. Turnout and Social Contact

#### 3.7.1. ‘Best Life’ Scenario

Within this theme social contact and the human-horse relationship were perceived as reflecting the conditions which horses living the ‘best life’ should have available to them. ‘Keeping the horses’ lives as natural as possible’ was seen as important. What constitutes ‘natural’ for a racehorse may be difficult to define but many of the participants saw turnout as an integral part of welfare outlined by the need to ‘turn out every day, elements, space/grass/outdoors/being able to buck and kick’, with a ‘choice of access to outside space’ and ‘paddock living with access to shelter by choice’.

In terms of social contact turning horses out together led to some disagreement. It was suggested turnout would give ‘wider access to social companions, group turnout (appropriately managed)’, although it was thought that horses should not be turned out in groups but could have ‘adjoining paddocks, groom but not kick’. 

#### 3.7.2. Minimum Welfare Standards Scenario

The converse of this was, in general, not seen as important for a horse kept under minimum welfare standards. Turnout was much less frequently mentioned with a ‘degree of turnout for same period each day, with/without grazing’. Turnout was also perceived as an area that could also be classed as a challenge to welfare when horses are turned out ‘whatever the weather conditions, even when it’s too hot and no care’. Social contact for the horse wasn’t seen as overly relevant in the minimum welfare scenario for the horse either, they were perceived as getting ‘visual access to other horses’.

### 3.8. Staff Management and Education

#### 3.8.1. ‘Best Life’ Scenario

Experience and knowledge were perceived as important factors, having, ‘experienced knowledgeable staff that will listen’ and ‘skilled staff in handling and riding’. 

#### 3.8.2. Minimum Welfare Standards Scenario

‘Caring and capable staff’ and ‘competent knowledgeable staff/trainer’ were still required when minimum welfare standards are employed. In contrast, some of the comments may be better classed as challenges as they imply a standard lower than the minimum, ‘staff not consistent and don’t notice if the horse is unwell or off colour’ and ‘more education of those involved in racing’. Some trainers were perceived as not having the experience and knowledge needed to train, a licence given on ‘... who you know, how much money you have’, rather than ‘[having an] understanding of racing/training education’, a case of nepotism and inequity versus fairness and ability. The trainers’ perceived lack of knowledge was viewed as being potentially detrimental to welfare and as such, better classed as a challenge.

### 3.9. Perceived Challenges to Welfare

A total of 68 welfare challenges were identified by participants. Health was again closely aligned to welfare but staff management and education moved from eight to second when ranked as a challenge.

#### 3.9.1. Health

Areas targeted were, for example, ‘the over use of veterinary interventions’ and ‘soundness’. On-going health issues included ‘repetitive injuries’, ‘chronic disease monitoring (including dentistry and veterinary aspects)’, ‘pain management’ (stated as more important than management issues) and ‘a greater robustness of substance analysis’.

Husbandry factors were perceived as impacting on health and such created challenges to welfare. These were ‘travelling horses without hay’ (contested by other participants because of the effect of spores and dust on the horse’s respiratory system in a confined area) together with ‘the withdrawal of hay for 24 hours before a horse ran’.

The thoroughbred sales were pinpointed as challenging welfare. Suggestions such as ‘horses were given too many endoscopic examinations at the sales’.

#### 3.9.2. Staff Management and Education. Ranked 2nd

‘The lack of knowledgeable experienced staff’ and ‘bad trainers or incompetent trainers’ were seen a challenge. As the perception of trainers’ education was deemed inadequate, it was suggested that ‘they should be made to take a trainer’s exam’ and ‘employment relations were seen as poor in some training yards and so should be investigated’. 

#### 3.9.3. Daily Routine and Monitoring 

‘The [lack of time] time spent out exercising horses’ as well as ‘the lack of staff to look after the horses in the yard when other staff are off racing’. More generally in the yard, there was a perceived ‘poor standard of care around horse husbandry when the stable routine is compromised through lack of staff’.

#### 3.9.4. Training, Exercise and Recovery

Within this category perceived challenges covered a horse in training from the beginning of her or his career to the care provided immediately after a run. ‘Horses were not given time when they were broken in’ and ‘they entered training at too young an age and weren’t given time to adapt physically and mentally to the training’. 

‘The lack of recovery time some horses were given after racing’ was highlighted as it was thought that when a horse’s race day experience is rushed, and the horse has not fully recovered before travelling home ‘their racing experience can affect their experiences the next time. Horses which then don’t perform as well are labelled “ungenuine’’. 

#### 3.9.5. Physical Comfort/Living Environment

It was thought ‘the level of health being affected by yard hygiene’ and ‘the incidence of low-grade bacterial infections’ could affect welfare.

#### 3.9.6. Turnout and Social Contact 

‘Meeting mental needs and social behaviour’ and ‘mental attitude and happiness’ together with the ‘the lack of opportunities to express normal behaviour as well as socialising’ were problems which needed solving. 

#### 3.9.7. Policy and Procedures

‘Running horses over the wrong distance’, for instance running a horse over a mile when the horse needs a mile and a half was seen as detrimental to the horse. 

#### 3.9.8. Other Areas

Owners and breeders were singled out, ‘being a horse who was owned by an owner who did not understand racing’ was a challenge to welfare and ‘a welfare issue when owners grab the horse if their horse has won’. The horse does not know them and can be startled. The motives of thoroughbred horse breeders were questioned, ‘horses were bred for performance rather than breeding from ‘tough’ horses’, that is, horses that had proved themselves without considerable amounts of veterinary intervention to keep them racing.

#### 3.9.9. Feeding

It was thought that ‘there was a lack of relevant feeding’ so potentially compromising welfare.

### 3.10. Innovative and Uncommon Practices

Exercise three asked participants to identify any innovative or uncommon practices they had witnessed or knew of. The statements have been analysed and ranked using the eight identified themes.

Ten of the 13 statements suggested as innovative or uncommon practices (see Table 3) are veterinary related, for example, ‘advances in veterinary care’, ‘frequent expert monitoring of lameness’, and as such would come under the ‘best life’ scenario in that they describe practices deigned to keep a horse in training fit and healthy. None of the statements were linked to minimum welfare standards. 

There were no statements that corresponded to daily routine and monitoring, feeding, staff management and education and policy and procedures.

## 4. Discussion

The aim of the study was to gain an understanding of racehorse welfare as perceived by racing industry stakeholders. Gaining access to field sites and participants can often be one of the most challenging parts of conducting research especially within an industry that has its own culture, language, customs and beliefs, such as the racing industry. Most groups tend to have things they do not want others to know about, especially non-members of that group; having a researcher who was part of the ‘racing tribe’ [25] and who like Cassidy [26] and Butler [27] could draw upon their biographical engagement in a specific social world facilitated access to the participants. 

The six participants from the Equine Referral Hospital, whilst not directly involved with racing on the racecourse were often dealing with injuries and conditions associated with horses in training and so were well placed to suggest some of the factors that would create the ‘best life’ and what the minimum welfare standards could be for a horse in training. Of the other three vets two worked as veterinary officers for the British Horseracing Authority at the racecourse whilst the third vet worked at specific racecourses, dealing with immediate traumas and also in a large successful flat yard in Newmarket. All three participants could draw on their experiences at the ‘coal face’ when discussing racehorse welfare.

However, what must be borne in mind is that the production of knowledge on racehorse welfare will have been influenced, shaped and conditioned by the subjectivity and positionality of many of the participants and will, in turn, shape the nature of the data collected [28]. Participants were drawn directly from or were closely involved with the racing field and generally accepted the values, practices and beliefs of racing, which tend to be viewed as inherently true and necessary, even though these values and beliefs may be arbitrary and contingent. 

Such views, in part, heightened some of the limitations that are associated with focus groups, that is recruiting participants [29]. Some participants thought assessing racehorse welfare a waste of time and so were not prepared to engage with the project when contacted to see if they would like to attend, a stance reiterated by participants when they mentioned their attendance to some of their clients. Arranging some of the groups was itself challenging in that, in one instance, the chairperson of one of the regional trainer’s organisations, the ‘gatekeeper’, that is the person that stands between the data collector and potential participants [30] was said to be ‘ill’ which meant we could not use his support of the study. This meant access to other members of the trainer’s organisation was much harder to achieve.

Non-attendance was also an issue; the onus to attend is on the participant and in some locations, emailing and phoning and personal visits led to ‘promises’ to attend, promises which were then broken. This in turn reduced anticipated numbers and groups although robust discussions were still had.

Using a snowballing technique worked well in that D.B. was able to use her contacts initially to recruit participants. However it did mean that each participant was recruiting the next participant thus creating a self-selected group using shared frames of meaning to present their opinions and experiences. This can have drawbacks in the level of agreement and consensus that is reached and may reflect the nature of the racing industry in that welfare is generally perceived as good [31]. In one group especially, where attendance was lower than expected due to non-attendance participants were initially ‘in awe’ of one outspoken participant. Nevertheless having similar frames of reference meant they were able, at times, to challenge and defend their opinions on contentious topics such as turning horses out, and turning out in groups.

Any welfare assessments will need to be based on good animal welfare science as well as making a reasonable fit with the major value positions as to what constitutes the ‘best life’ for racehorses in training. Nonetheless, the racing industry in Britain is perceived as having a higher risk in terms of welfare concerns than other equine sectors such as those seen in leisure horses or in show jumping [32] and media coverage of racehorse deaths at prestigious race meetings have not allayed the fears of some of the non-racing public that racing is cruel, does compromise welfare, and so should be banned. 

Participants were able to identify more elements that would influence a ‘best life’ scenario, (56% of the statements) compared to the minimum. It is difficult to pinpoint exactly why this might be although it may relate back to the desire to individualise each horse with regard to their needs so within the ‘best life’ scenario there will be more variable factors that can be considered. A ‘one size fits all’ approach is more homogenised and as such would not be expected to generate additional factors. 

In each of the eight themes two recurrent strands, contributory factors associated with health and the human-horse relationship provide the narrative which participants have created to construct their perception of a ‘best life’ where the horse is treated as an individual compared to one predicated upon minimum standards, a ‘one size fits all’ approach. Focus group participants made a comprehensive distinction between what they perceived as the ‘best life’ when compared to one where minimum standards are utilised in all themes except for policy and procedures where practices suggested were also described as standard procedures.

A trainer’s approach to training will be, to a large extent, determined by value-based ideas about what elements are important for horses to have the ‘best life’ where ‘best’ equates to having horses fit to compete to the best of the horses ability which may be why health was seen as the most important overall factor. From this perspective it is important, therefore, to make sure the horse has all of his or her mental and physical needs met. Within the ‘best life’ scenario veterinary care was closely aligned to health. This equated to evidence based treatment, for example, where ‘wind operations’, were performed only in response to a problem [33]. Other factors included having a vet immediately available and horses gastroscoped twice a year for equine gastric ulcer syndrome (EGUS). Participants views that ‘gastroscopy should be performed at least once a year’ as a minimum welfare standard are substantiated by research indicating that the prevalence of the disease has been reported as high as 100% in racehorses. In general the prevalence of EGUS is around 90% to 93% in racehorses in active training [34,35]. Causes are multi-factorial but diets high in soluble carbohydrates, the length of time in training, psychological stress and constant stabling are said to significantly increase the risk of horses developing EGUS [36]. In a ‘best life’ scenario it was suggested horses would have no gastric ulcers. This may be possible to achieve given the feeding methods that were described, for instance, access to *ad libitum* forage fed to the individual’s needs which would be monitored by staff, specialist concentrate feed, tailored to each horse and their level of work, fed three to four times a day and pasture turnout. Such an approach should mean there would rarely be an absence of food in the horse’s stomach buffering the presence of acid which is always present in the glandular region of the stomach [37]. 

In a minimum welfare scenario feeding was perceived as quite basic, with little variation in the type of feed fed, horses would only be fed concentrate feed three times a day where forage may be severely restricted or only given twice a day. 

Turnout was the most contentious issue discussed by participants. At pasture, horses spend most of their day with their heads down close to the ground, grazing and moving [38]. Although the evidence supporting grazing is conflicting, continuous access to good quality grass pasture was viewed as ideal in the prevention of gastric ulcers and in the beneficial stretching of the soft tissues along a horse’s back [39,40]. Some participants saw turnout as part of a ‘best life’, enhancing social contact through regular turnout with wider access to social companions, ‘keeping the horses’ lives as natural as possible’ by providing grazing, physical space and other pleasures such as being in the sun as emphasised in the Code of Practice for the Welfare of Horses, Ponies, Donkeys and their Hybrids [5]. Some participants advocated turnout in individual paddocks with some socialisation with their next door neighbour, which as Henderson [41] illustrates can allow horses to groom each other over a fence opening up the possibility of pairing them up together. Others however, thought the risk of injury, especially with young horses outweighed the benefits that turnout can give [42]. Turnout was sometimes identified within a minimum welfare scenario but with a more inflexible regime where horses were turned out regardless of the weather, for instance, extreme weather conditions where little shelter was available. 

In some yards turnout will be part of the routines which are bound to the daily management practices that are carried out in a racing yard from feeding, mucking and riding out which, apart from riding is repeated again in the afternoon [27]. Horses living the ‘best life’ were perceived as having an established routine which could be changed if it did not suit the individual horse. This may include, for instance, changing the time when the horse is ridden out, giving them a different stable and making sure they have the same person looking after them and riding them out on a daily basis. Horses kept under minimum welfare standards would tend to follow a similar logistical routine although participants thought that there would be little variation between how individual horses were treated with few staff available who would not have the time to spend with each horse. In terms of minimum standards and ‘best life’ physical comfort was similar in that horses should have enough bedding or a suitable non-slip surface to allow them to roll with a large enough stable for the size of the horses. Where there was a wide variation between the two scenarios was in horses’ living environment.

A range of environments in which horses can be kept was seen as an important aspect in providing horses with the ‘best life’ they could have whilst in training, for instance, ‘rabbit hutch’ stables with corrals at the front and front and back openings where the horse could look out. The key difference was for a horse living the ‘best life’ was that their environment should reflect the needs of the individual horse by providing, where possible, a variety of stabling design available on the premises. Participants identified similar types of stable design as innovative practices where horses have some social contact with each other. Where it was not possible to alter stable design panoramic images or visual images of a horse on the back of the stable wall were reported as working well in creating a calmer internal environment [43]. Studies on stabled horses found that stabled horses are motivated by the possibility of interacting with other familiar horses [44], a factor that was identified by participants. Having grills between stables so horses could sniff and see their neighbour and open sided stables were perceived as enhancing social contact as well as reducing or preventing stereotypical behaviour and lowering stress levels in horses [45,46].

Participants identified that treating each horse in training as an individual requires team work. A team approach by health professionals and changes in management can strengthen and improve overall health and career longevity as well as instigating a more proactive attitude as participants are intimating [47,48]. Examples of such an approach were a collaborative preventative health care plan for individual horses which included teamwork, the team made up of farriers, dentists, vets and paraprofessionals such as physiotherapists. Although there is a lack of scientific investigation into the use and efficiency of such therapies, many horse owners have been found to have used some form of perceived therapeutic treatment for their horse, the most common being massage [49,50].

Some of the practices suggested as innovative or uncommon practices in terms of health were also practices suggested as part of a ‘best life’ scenario, for instance, regular dentistry care and endoscoping, the use of spas and para-professionals. All of the practices outlined come at a monetary cost and have a time element attached to them. For a less established trainer the cost to their business and the costs which are passed on to their owners may make using additional treatments and alternative routines too financially prohibitive for them to integrate as part of their health care routine.

In terms of human input in creating a healthy ‘best life’ environment, strong leadership and management were viewed as important in enabling a co-ordinated approach to the individual assessment of each horse taking into account their work and care. In order to maintain levels of care it was suggested that, as a way of expanding and enhancing a trainer’s knowledge and skills trainers should have to undertake some form of continuous professional development and for staff, rider coaching and staff development [51]. Nonetheless health also ranked first in terms of challenges, where participants outlined how the push to have horses fit to compete meant there was concern surrounding the overuse of veterinary interventions such as medication of joints and wind operations.

Minimum standards for health were described as ‘basic’ where welfare is ‘paramount’ but veterinary care is used only when needed, a ‘fire brigade’ [52] approach where the vet or farrier only called in at the last minute. Participant’s views of health when minimum welfare standards are applied are in keeping with the findings of McGowan et al. [53] of standardised practices with a lack of integration and limited communication in the way equine health care is sometimes approached. 

Maintaining an individual approach to each horse was emphasised within the training, exercise and recovery theme. Cross training, for instance, elementary schooling was suggested in a ‘best life’ scenario as part of a varied exercise routine. It appears to be protective against days lost to injury, more because it reduces the risk of repetitive overload injuries and may also maintain a horse’s motivation for work [54]. However it needs experienced knowledgeable staff to carry out the different forms of training plus the time available if it is to be integrated into a daily training and exercise programme. 

From what participants have indicated, to treat each horse as an individual requires ‘well trained,’ ‘knowledgeable’ staff, just as with minimum standards, but for the ‘best life’ the horse-human relationship is built upon ‘experience’, in contrast to a minimum welfare scenario. ‘Care’ was viewed as important in both scenarios although it was emphasised more strongly from a ‘best life’ perspective. Staffing levels and the importance of ‘matching the rider to the horse’, were flagged up as part of a ‘best life’ scenario with regard to daily routine and monitoring and in training, exercise and recovery. As Hausberger et al. [55] found the horse–human relationship is most beneficial for both parties when the relationship is built upon the basis of a succession of interactions and the development and maintenance of a really positive relationship, perceived by participants in our study as ‘a consistency of routine and carer’ and ‘the same person looking after and riding the same horse every day if they get on’, a characteristic outlined by Williams and Tabor, [56] within the wider equine industry. 

Statements included within the ‘best life’ scenario illustrate the need for the horse to be cared for as an individual, more easily achieved by higher staffing levels. Staff look after fewer horses thus seen as creating a positive horse-human contact leading to a potentially higher level of care and observation of each horse. An almost identical scenario with regard to welfare was recorded by Boivin et al. [57] with farm animals. If the human–animal relationship was positive welfare and productivity was less likely to be compromised although as they highlight, the trend in agriculture as in the racing industry, is for stock people to care for more animals. The ‘best life’ however is harder to achieve if staffing levels are low, there were not enough experienced staff present and staff are away taking horses racing. This was a factor highlighted by participants as a challenge, due to the ongoing staff shortage and the perceived lack of experienced knowledgeable staff. Nevertheless, staff shortages are not a new phenomenon for the racing industry [27] and changes to UK immigration policy may further increase the estimated shortfall of 1000 stable staff [58].

One of the consequences of a smaller pool of staff within the industry and the reliance on casual staff is that some yards struggle to exercise the horses they have in training on a daily basis, either because there are not enough experienced staff to ride the more excitable horses or there are simply not enough riders or experienced yard staff on the ground. Whilst horse walkers can be used and were included as part of a minimum standards scenario they are no substitute for a horse being correctly ridden in terms of weight bearing and strengthening of the musculoskeletal frame, thus avoiding many of the injuries that may otherwise be sustained [59].

Across all three exercises health came out as the common denominator in participant’s perceptions of racehorse welfare. Good health was indicative of a ‘best life’, thus producing horses fit to race yet concomitantly it was part of a minimum welfare scenario through the overuse of medication to keep a horse racing, also identified as a challenge to welfare [38]. Maintaining a high level of health care requires knowledgeable, experienced trainers and staff thus maintaining the quality of the horse-human relationship. Nonetheless the lack of knowledgeable staff and poor employment relations identified as challenges may exacerbate some of the factors identified as part of a minimum welfare scenario which, in turn, can impact on welfare, such as inadequate staffing levels in yards when other staff are away racing. 

The understanding gained through this study will be used, in conjunction with the scientific animal welfare literature, to develop a welfare assessment protocol for racehorses in training. All the elements identified as affecting welfare, in either the BL or MWS scenarios, will be incorporated as far as possible, in a pilot protocol that will undergo feasibility testing and further stakeholder consultation. The purpose of using the two MWS and BL scenarios was not so much to identify an ‘acceptable’ threshold of welfare but rather to encourage wide-ranging thinking about all the elements affecting welfare. The protocol will be used for assessing yards, and therefore a group of horses, but the assessment will include observations of individual horses, as well as information about the group as a whole. The mechanism for use of the protocol, especially in any regulatory capacity, for example linked to granting a trainer’s licence, is as yet undetermined.

## 5. Conclusions

The aim of this study was to explore the perceptions of racehorse industry stakeholders in regard to racehorse welfare. A trainer relies on the good performance of his or her horses to advertise their expertise and skill in training racehorses. In an ideal world their horses would be living the ‘best life’ as identified by participants, with each horse treated as an individual, whose health and well being was monitored and assessed by an experienced and caring team of staff, including vets and para-professionals. Each horse would have a training plan mapped to their needs and fed according to the level of work they undertook. They would be kept in an environment that suited their temperament where turnout in groups was a daily occurrence. A ‘best life’ scenario such as this relies on the horse-human relationship for it to be implemented, monitored and changed if it is not working. However, as participants discussed staffing levels are a challenge, making it difficult to provide the type of care and attention participants would like to give resulting in compromises being made. Given the staff shortages inherent within the racing industry it may be difficult to provide a ‘best life’ scenario, trainers may have to implement a mix of both scenarios if they are to keep their horses fit and healthy. 

Health and health related factors were perceived as having positive and as well as negative outcomes for horses’ welfare. A healthy horse has the potential to perform to the best of his or her ability yet health was negatively aligned to welfare when perceived as a challenge. It is difficult to draw a line between what participants perceive as factors for a ‘best life’ and those which are either minimum welfare standards, or are verging on negative welfare outcomes. This is illustrated by participants’ understanding of veterinary interventions used to keep a horse racing as well as husbandry factors such as reduced forage intake. Both were perceived as being detrimental to welfare, yet advances in treatment, improved techniques and diagnostic methods were welcomed as innovative. The data this study has produced, when considered alongside practical animal welfare science, will be used to develop a holistic racehorse welfare assessment protocol—a first for the racing industry.

## Figures and Tables

**Table 1 animals-09-00134-t001:** Demographic breakdown of focus group participants by occupation and location.

Main Stakeholder Groups	Midlands	Newmarket	Middleham	Lambourn	VS^1^	London	Cotswolds	Total
Trainers	1	0	3	4	0	0	0	**8**
Assistant trainers	1	2	0	1	0	0	0	**4**
Stable staff	2	1	2	2	0	0	0	**7**
Owners	1	2	0	0	0	0	0	**3**
Vets	0	0	0	0	5	2	0	**7**
Animal Charity employees	0	0	0	0	0	4	0	**4**
Racing Welfare	0	1	1	0	0	0	0	**2**
Paraprofessionals	1	0	0	0	0	0	1	**2**
BHA veterinary officers	0	0	0	0	0	0	1	**1**
BHA inspectors	0	0	0	0	0	0	2	**2**
Other	0	1	0	1	0	0	0	**2**
**Total**	**6**	**7**	**6**	**8**	**5**	**6**	**4**	**42**

^1^ Veterinary School

**Table 2 animals-09-00134-t002:** Number of statements per theme and associated frequency ranking.

Themes	Minimum Welfare Standards	Standard Procedures	‘Best Life’	Totals	Proportion of Statements Per Theme	Ranking
Health	24	0	34	58	24%	1
Training, exercise and recovery	21	0	22	43	18%	2
Physical comfort/living environment	16	0	24	40	16%	3
Feeding	15	0	15	30	12%	4
Daily routine and monitoring	13	0	13	26	11%	5
Policy and procedures	0	8	8	16	6.5%	6
Turnout and social contact	4	0	12	16	6.5%	6
Staff management and education	6	0	9	15	6%	8
Total	99	8	137	244	100%	

**Table 3 animals-09-00134-t003:** Innovative or uncommon practices to improve welfare that stakeholders were aware of (27 statements in total).

Themes	Innovative or uncommon practices
Health (13 statements)	Advances in veterinary care Understanding drugs and what they can be used for Use of sedatives Regular dentistry care Hormone treatment for fillies Veterinary practices Blood monitoring, scoping, X-rays, scan Frequent expert monitoring of lameness Stem cell research, electrotherapy. Improved treatments for kissing spines Climate control in horseboxes Spas Water treadmills
Physical comfort/living environment (5 statements)	Panoramic views in stables Full tree saddles instead of half tree saddles Use of tack such as bungees, ‘pessoas’, elasticated nosebands Improved saddle design Gorse/wooden blocks in stables
Training, exercise and recovery (5 statements)	All weather surfaces Wexford gallops (deep circular sand gallops) Whipless racing Indoor facilities (covered rides) Better ground at racecourses
Turnout and social contact (4 statements)	Temporary turnout paddocks at racecourses Other species, eg, sheep, as a companion Crescent shaped yard where horses can ‘talk’ and see each other with corralled area at the front. Rabbit hutch’ boxes with small corralled area at the front

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
