# Peer review of "Living the ‘Best Life’ or ‘One Size Fits All’—Stakeholder Perceptions of Racehorse Welfare"

_animals, 2019, doi:10.3390/ani9040134_

Round 1

Reviewer 1 Report

Article: Living the “best life” or  “one size fits all” – Stakeholder perceptions of racehorse welfare.

In this manuscript a series of discussions done through focus groups were done with different stakeholders from the horseracing industry in order to understand their perception of racehorse welfare. This considering “best life” and “minimum standards” for the horses. In general, I think this is an interesting article that tackles a subject poorly reported in the literature.

I have some general comments about the structure of the article.

In the introduction section I believe a brief description of the management of race horses in the UK would benefit the article. Do these horse live at the track? Are they transported from farms to the race track every time they have a race? Feeding scheme?  Are they kept mainly in stables? Do they have access to free exercise?. This because the management of race horses varies across different countries and it could allow better understanding of some of the stakeholders perceptions.

Materials and Methods:

The statements were analyzed through thematic analysis, I think using text mining would allow a better description of the results, author would be able to extract clusters and see if there is any type of groups according to statements, for example between territories or type of stakeholder. It would be interesting to see if trainers, assistant trainers, stable staff and owners share a common view that might differ from vets, animal charities, racing welfare and finally from those that are inspectors or officers.

Results:

This section is a little difficult to follow due to the large amount of statements that are provided.

Line 185-190:  would fit better in the discussion section.

Table 3: number of statements for minimum welfare standards are missing for para-professionals and Additional therapy.

Line 332-333 I think the sentence needs to be structured differently  (the two scenarios was a money no object approach…?)

Maybe in the results section instead of given almost all statements within the text, just one or two good examples would make the manuscript easier to follow. The complete list of statements by theme could be provided as a table in the supplementary material section.

Discussion:

Line 610: please mention this other sectors

Line 631: “wind operation” is a colloquial term and used in all countries, it should be defined as surgical procedures associated to upper airways in horses (larynx, soft palate).

L633-637: include some references about the magnitude of the gastric ulcers problem in the UK

L643-644. Do you have a reference for this statement about turn out and gastric ulcers?

I think some more references should be provided in the discussion section on how the perceptions of stakeholder relate to actual welfare issues in the racing industry, I understand there is not much information on attitudes and perceptions, but there is literature on some of the common welfare and health issues associated to these horses.

What should be aimed at in a welfare protocol? To accomplish the best life scenario or the minimum standards? Should these types of welfare protocols assess welfare at an individual or group level? I think something should be commented about this.

Author Response

Manuscript ID: animals-468987

Title: Living the ‘best life’ or ‘one size fits all’- Stakeholder perspectives of racehorse welfare

Editor’s decision: major revisions

Responses to Reviewers comments

We would like to take this opportunity to thank the three reviewers for their constructive comments. We think that our manuscript is stronger as a result of the revisions made and hope that you are in agreement.

I have added the suggested changes to the manuscript in red italics and have included any additional text in the rebuttal. Where new references have been added these are shown as [xx] in the text and I have noted the page number, again, within the rebuttal.

I have also added in comments to the manuscript using track changes.

Reviewer 1.

Introduction

In the introduction section I believe a brief description of the management of race horses in the UK would benefit the article. Do these horse live at the track? Are they transported from farms to the race track every time they have a race? Feeding scheme?  Are they kept mainly in stables? Do they have access to free exercise?. This because the management of race horses varies across different countries and it could allow better understanding of some of the stakeholders perceptions.

I have added In 2018 there were just under 600 licensed racehorse trainers training 23,599 racehorses per annum,  who ran on average 5.71 times on the flat, 4.22 times in National Hunt races [3].  Racehorse training yards are located over the length and breadth of Britain, and vary in size from less than ten horses in training to over 250 [4]. Each trainer will have their own methods of getting horses fit which, to some degree, will be dependent on the training facilities they have.  Feeding methods and the types of concentrates and forage fed will also vary but usually horses will be fed three to four times a day.  The use of turnout for racehorses, either individually or in groups is used by some trainers although it is not a universal practice mainly on grounds of risk and the ground trainers have available. Horses are transported to race meetings by lorry then stabled in racecourse stabling where strict procedures are in place limiting access to the stables to individuals registered by the British Horseracing Authority.  Trainers are reliant on the staff they employ to carry out the day to day and an ongoing staff shortage in Britain [5] has resulted in a working practices and the division of labour of yards has changed. The role of the racing groom was once typically a composite one, the elements of which have now been separated out into skilled and less skilled roles within the occupational hierarchy of a yard

Material and methods

The statements were analyzed through thematic analysis, I think using text mining would allow a better description of the results, author would be able to extract clusters and see if there is any type of groups according to statements, for example between territories or type of stakeholder. It would be interesting to see if trainers, assistant trainers, stable staff and owners share a common view that might differ from vets, animal charities, racing welfare and finally from those that are inspectors or officers.

I re-analysed the statements as they were produced at individual focus groups.  In general most of the participants held common views, for example, stable staff mentioned the overuse of veterinary interventions, that trainers don’t always listen as did some of the vets. I have however added a paragraph in the discussion which draws the reader’s attention to the fact that the participants are part of a racing field where individuals develop the attitudes and dispositions of that cultural field as well as shaping the ways in which those individuals engage in practices.

 However what must be borne in mind that the production of knowledge on racehorse welfare will have been influenced, shaped and conditioned by the subjectivity and positionality of the participants [28]. Participants were drawn from the racing field and on the whole accepted the values, practices and beliefs of racing, which tend to be viewed as inherently true and necessary, even though these values and beliefs may be arbitrary and contingent.

Results

This section is a little difficult to follow due to the large amount of statements that are provided.

Yes! See last suggestion in this section of comments and suggestions.

Line 185-190:  would fit better in the discussion section.

I removed it when trimming down the results section

Table 3: number of statements for minimum welfare standards are missing for para-professionals and Additional therapy. These were added, 0 and 0 and the tables are now in supplementary materials.

Line 332-333 I think the sentence needs to be structured differently (the two scenarios was a money no object approach…?).

I have replaced this with line 280, ‘was an economic one’

Maybe in the results section instead of given almost all statements within the text, just one or two good examples would make the manuscript easier to follow. The complete list of statements by theme could be provided as a table in the supplementary material section.

I have removed many of the statements as suggested just using ones which participants had highlighted as important factors with regard to welfare.

Discussion

Line 610: please mention this other sectors

Line 480. Added ‘seen in leisure horses and showjumping’

Line 631: “wind operation” is a colloquial term and used in all countries, it should be defined as surgical procedures associated to upper airways in horses (larynx, soft palate).

Added in, ‘surgical procedures associated with upper airway obstructions  in horses’

L633-637: include some references about the magnitude of the gastric ulcers problem in the UK

Lines 510-510

I have added,  Participants views that gastroscopy should be performed at least once a year as a minimum welfare standard are substantiated by research indicating that the prevalence of the disease has been reported as high as 100% in racehorses. In general the prevalence of EGUS is around 90% to 93% in racehorses in active training [34, 35]. 

The references I have used are, [34] Nadeau, J.A.; Andrews, F.M. Equine gastric ulcer syndrome: The continuing conundrum.  Eq.Vet.J  2009, 41, 611-15.  Available online:

https://www.researchgate.net/publication/40021059_Equine_gastric_ulcer_syndrome_The_continuing_conundrum (Accessed 15/03/19)

[35]. Bell, J.W.; Mogg, T.D.; Kingston, J.K. Equine gastric syndrome in adult horses: a review. N.Z.Vet J 2007, 55, 1-12. https://doi.org/10.1080/00480169.2007.36728

L643-644. Do you have a reference for this statement about turn out and gastric ulcers?

Lines 520-524.   I have added, Turnout was the most contentious issue discussed by participants. At pasture, horses spend most of their day with their heads down close to the ground, grazing and moving [38]. Although the evidence supporting grazing is conflicting continuous access to good quality grass pasture was viewed as ideal in the prevention of gastric ulcers and in the beneficial stretching of the soft tissues along a horse’s back  [39, 40].

The reference is, [39] Sykes, B.W.; Hewetson, M.; Tamzali, Y. European College of Equine internal Medicine Consensus Statement-Equine Gastric Ulcer Syndrome in Adult Horses. J Vet Int. Med 2015, 29, 1288-1299

I think some more references should be provided in the discussion section on how the perceptions of stakeholder relate to actual welfare issues in the racing industry, I understand there is not much information on attitudes and perceptions, but there is literature on some of the common welfare and health issues associated to these horses.

I have added a number of new references that refer to some of the common welfare and health issues. I have referenced these in the text in red italics [xx].

What should be aimed at in a welfare protocol? To accomplish the best life scenario or the minimum standards? Should these types of welfare protocols assess welfare at an individual or group level? I think something should be commented about this.

We have added,

The understanding gained through this study will be used, in conjunction with the scientific animal welfare literature, to develop a welfare assessment protocol for racehorses in training. All the elements identified as affecting welfare, in either the BL or MWS scenarios, will be incorporated as far as possible, in a pilot protocol that will undergo feasibility testing and further stakeholder consultation. The purpose of using the two MWS and BL scenarios was not so much to identify an ‘acceptable’ threshold of welfare but rather to encourage wide-ranging thinking about all the elements[C1]  affecting welfare. The protocol will be used for assessing yards, and therefore a group of horses, but the assessment will include observations of individual horses, as well as information about the group as a whole. The mechanism for use of the protocol, especially in any regulatory capacity, for example linked to granting a trainer’s licence, is as yet undetermined.

 [C1]Reviewer 1. Paragraph on aims the racehorse assessment protocol.

Reviewer 2 Report

This manuscript reports the findings of focus groups conducted to identify individual perceptions of factors affecting Thoroughbred racehorse welfare. This is an important issue which, as the authors rightly indicate, is difficult to assess using the currently available instruments. Therefore, there is merit in the research being reported. The method chosen to conduct the study is associated with inherent problems in the focus group approach, in particular in the recruitment of subjects. The authors should address them more explicitly in the discussion.Focus groups are a valid way of collecting the information the authors set out to collect and the method is therefore appropriate. The method used to analyse the data appears to be appropriate.

The results and discussion sections are both quite long. The manuscript would benefit from editing to reduce the narrative, especially where it is discussing a response made by a single participant. The significance of these responses is debatable and I feel that they add little to the key outcome reported in the manuscript, which is the identification of the important themes underlying people's perception of welfare. It may be possible to include much of this information as supplementary material. The concerns expressed may reflect the respondent's unique experiences which, while important, may not be of particular relevance to the overall subject of welfare.

In summary, I believe that the manuscript has the potential to make a worthwhile contribution to the discussion around racehorse welfare. I suggest that the authors investigate ways in which the word count can be reduced. I think it is very important that the authors make clear the limitations of this study, in particular the potential problems surrounding the focus group method, and the fact that the results are subjective, and are likely to be influenced by participants' inherent biases.

Author Response

Manuscript ID: animals-468987

Title: Living the ‘best life’ or ‘one size fits all’- Stakeholder perspectives of racehorse welfare

Editor’s decision: major revisions

Responses to Reviewers comments

We would like to take this opportunity to thank the three reviewers for their constructive comments. We think that our manuscript is stronger as a result of the revisions made and hope that you are in agreement.

I have added the suggested changes to the manuscript in red italics and have included any additional text in the rebuttal. Where new references have been added these are shown as [xx] in the text and I have noted the page number, again, within the rebuttal.

I have also added in comments to the manuscript using track changes.

Reviewer 2

This manuscript reports the findings of focus groups conducted to identify individual perceptions of factors affecting Thoroughbred racehorse welfare. This is an important issue which, as the authors rightly indicate, is difficult to assess using the currently available instruments. Therefore, there is merit in the research being reported. The method chosen to conduct the study is associated with inherent problems in the focus group approach, in particular in the recruitment of subjects. The authors should address them more explicitly in the discussion.

Lines 459-486

However what must be borne in mind that the production of knowledge on racehorse welfare will have been influenced, shaped and conditioned by the subjectivity and positionality of the many of the participants which will, in turn, shape the nature of the data collected [28]. Participants were drawn directly from, or were closely involved with the racing field and generally accepted the values, practices and beliefs of racing, which tend to be viewed as inherently true and necessary, even though these values and beliefs may be arbitrary and contingent[C1] .   Such views, in part, heightened some of the limitations that are associated with focus groups, that is recruiting participants [29].  Some participants thought assessing racehorse welfare a waste of time and so were not prepared to engage with the project when contacted to see if they would like to attend, a stance reiterated by participants when they mentioned their attendance to some of their clients.

Arranging some of the groups was itself challenging in that, in one instance, the chairperson of one of the regional trainer’s organisations, the ‘gatekeeper’, that is the person that stands between the data collector and potential participants [30] was said to be ‘ill’ which meant we could not use his support of the study which meant access to other members of the trainer’s organization was much harder to achieve.

Non attendance was also an issue; the onus to attend is on the participant and in some locations, emailing and phoning and personal visits led to ‘promises’ to attend, promises which were then broken. This in turn reduced anticipated numbers and groups although robust discussions were still had.

Using a snowballing technique worked well in that D.B. was able to use her contacts initially to recruit participants. However it did mean that each participant was recruiting the next participant thus creating a self-selected group using shared frames of meaning to present their opinions and experiences. This can have drawbacks in the level of agreement and consensus that is reached and may reflect the nature of the racing industry in that welfare is generally perceived as good [31].  In one group especially, where attendance was lower than expected due to non-attendance participants were initially ‘in awe’ of one outspoken participant. Nevertheless having similar frames of reference meant they were able, at times, to challenge and defend their opinions on contentious topics such as turning horses out, and turning out in groups.

Focus groups are a valid way of collecting the information the authors set out to collect and the method is therefore appropriate. The method used to analyse the data appears to be appropriate.

Results and discussion

The results and discussion sections are both quite long. The manuscript would benefit from editing to reduce the narrative, especially where it is discussing a response made by a single participant. The significance of these responses is debatable and I feel that they add little to the key outcome reported in the manuscript, which is the identification of the important themes underlying people's perception of welfare. It may be possible to include much of this information as supplementary material. The concerns expressed may reflect the respondent's unique experiences which, while important, may not be of particular relevance to the overall subject of welfare.

I have edited all of results to reflect the main perceptions of participants. I have removed the tables into the supplementary materials section. I have added another table in the supplementary section that outlines the statements per theme.  The discussion had been edited and reflects the main outcomes as outlined in the results.

In summary, I believe that the manuscript has the potential to make a worthwhile contribution to the discussion around racehorse welfare. I suggest that the authors investigate ways in which the word count can be reduced. I think it is very important that the authors make clear the limitations of this study, in particular the potential problems surrounding the focus group method, and the fact that the results are subjective, and are likely to be influenced by participants' inherent biases.

I have reduced the word count considerably. Without the references the paper is now just over 8500 words.

In the response above (in red italics)I have included a paragraph on some the limitations we encountered when using focus groups. I have highlighted the subjective nature of the data collected

 [C1]Added to draw the reader’s attention to participant’s knowledge.

Reviewer 3 Report

The main problem with this study is that almost all the veterinarians were from a veterinary college.  This should be mentioned in the discussion because te opinions of track vets are probably very different from those of academics
 Were the horses under the care of these stakeholders flat or National hunt?
Results what are these percentages of, Please clarify for the reader
Track veterinarians should have been included
P 6 ‘industry experts with no incentive to present to the vet’ 
 not sure what that means.  If the chiropractor has seen the horse the vet won’t be called?
3.3.2 ‘Minimal basic bedding … limited shavings’   These statements seem to imply that horse should have less rather than more bedding Is that true?
104 “pastoral care of stable personnel” 
 is this religious support or more economic
243 what are wind operations?
378 one participant who
Define MWS and BL on first use of these initials also CPD
“racing clearance post stand-downs’”  What does that mean to non race horse reader
541 horse’s horse vs horses should be uniform throughout the sentence. Make all plural
628 horses’ abilities
644 need reference for turnout  and ulcers
404 ‘racing clearance post stand-downs’  don’t know what that means; for example what are stand downs
414 what is CPD  tables Don’t understand  how MWS can be zero  and Best life 7, but 3 are common to both.
493  where do the18 statements and 14 statements come from/
715 what is cross work and grid work? Flat work I assume means  galloping with no fences.
784 best of his ability
Tables Don’t understand  how MWS can be zero  and Best life 7, but 3 are common to both

Author Response

Manuscript ID: animals-468987

Title: Living the ‘best life’ or ‘one size fits all’- Stakeholder perspectives of racehorse welfare

Editor’s decision: major revisions

Responses to Reviewers comments

We would like to take this opportunity to thank the three reviewers for their constructive comments. We think that our manuscript is stronger as a result of the revisions made and hope that you are in agreement.

I have added the suggested changes to the manuscript in red italics and have included any additional text in the rebuttal. Where new references have been added these are shown as [xx] in the text and I have noted the page number, again, within the rebuttal.

I have also added in comments to the manuscript using track changes.

Reviewer 3

The main problem with this study is that almost all the veterinarians were from a veterinary college.  This should be mentioned in the discussion because te opinions of track vets are probably very different from those of academics 

We tried to include more track vets but logistically it proved impossible to get them altogether to hold a focus group with them. However the vets we did include, whilst some may be classified as ‘academics’ they do also work in racing yards and deal with injuries and conditions that are a direct result of being a horse in training. The two British Horseracing Authority vets were well positioned to discuss racehorse welfare as it is they who assess horses on the track and before they run, in the paddock. The third vet has international experience of working as a racehorse vet  as well as dealing with horses in training in a large successful flat yard. I have added in the following, as you suggest, in the discussion.

Lines 142-144. I have added in

Lines 451-458

The six participants from the Equine Referral Hospital, whilst not directly involved with racing on the racecourse were often dealing with injuries and conditions associated with horses in training and so were well placed to suggest some of the factors that would create the ‘best life’ and what the minimum welfare standards could be for a horse in training. Of the other three vets two worked as veterinary officers for the British Horseracing Authority at the racecourse whilst the third vet worked at specific racecourses, dealing with immediate traumas and also in a large succesful flat yard in Newmarket. All three participants could draw on their experiences at the ‘coal face’ when  discussing racehorse welfare.

Were the horses under the care of these stakeholders flat or National hunt?

They were from both codes. I have added,

Line 110. ‘Participants with experience in both National Hunt (jump) and flat racing were recruited...’

Results

Results what are these percentages of, Please clarify for the reader. 

They are the  Proportion of statements per theme’.  I have added this to table 2

Track veterinarians should have been included

We tried to include as many different vets as possible. Hopefully I have explained the background of the vets we had attend. (See rebuttal earlier, lines 451-458).

industry experts with no incentive to present to the vet’ not sure what that means.  If the chiropractor has seen the horse the vet won’t be called?

I have removed this line as it is quite ambiguous. What the participant was referring to was that in some cases the para-professionals used should be experienced and knowledgeable enough to be able to assess whether a vet is needed or not.

3.3.2 ‘Minimal basic bedding … limited shavings’   These statements seem to imply that horse should have less rather than more bedding Is that true?

I have clarified this point as the participant was referring to the fact that in some yards staff are limited to the amount of bedding they can use.

Line  274. A minimum welfare standards scenario was more generalised, ‘minimal basic bedding,’ ’limited shavings, (where staff are limited on the amount of bedding they can put in).

pastoral care of stable personnel” is this religious support or more economic

I have changed this to,

Line 127.  ‘support the workforce of British horseracing’. 

 what are wind operations?

I have changed this to,

Line 236. surgical procedures associated with upper airway obstructions in horses’

 one participant who

Removed from manuscript.

Define MWS and BL on first use of these initials also CPD

I have defined these within the manuscript.

Line 160-161 a flip chart paper divided into two sections marked ‘minimum welfare standards’ (MWS) and ‘best life’ (BL).

Line 327. CPD [continuous professional development]

“racing clearance post stand-downs’”  What does that mean to non race horse reader

Lines 317-319.  ...there should be, ‘an agreed set of interventions for racing clearance post stand-downs’.  What was implied was that when a horse has fallen or pulled up they should be prevented from running again until they had been checked by a racecourse vet using an agreed protocol...

horse’s horse vs horses should be uniform throughout the sentence. Make all plural

The line has been removed

628 horses’ abilities

Line removed

need reference for turnout  and ulcers

I have added the following references,

Lines 520-524.   I have added, Turnout was the most contentious issue discussed by participants. At pasture, horses spend most of their day with their heads down close to the ground, grazing and moving [38]. Although the evidence supporting grazing is conflicting continuous access to good quality grass pasture was viewed as ideal in the prevention of gastric ulcers and in the beneficial stretching of the soft tissues along a horse’s back  [39, 40].

The reference is, [39] Sykes, B.W.; Hewetson, M.; Tamzali, Y. European College of Equine internal Medicine Consensus Statement-Equine Gastric Ulcer Syndrome in Adult Horses. J Vet Int. Med  2015, 29, 1288-1299

racing clearance post stand-downs’  don’t know what that means; for example what are stand downs

Lines 317-319.  ...there should be, ‘an agreed set of interventions for racing clearance post stand-downs’.  What was implied was that when a horse has fallen or pulled up they should be prevented from running again until they had been checked by a racecourse vet using an agreed protocol...

what is CPD

Line 327. CPD [continuous professional development]

 Don’t understand  how MWS can be zero  and Best life 7, but 3 are common to both

Good point. I’ve changed the ‘common to both’ to ‘standard procedures’, which participants thought should be carried out as a matter of course.

Lines 195-6.  Eight statements relating to policy and procedures were allocated to standard procedures

where do the18 statements and 14 statements come from

These have been removed

what is cross work and grid work? Flat work I assume means  galloping with no fences.

Lines 240-243.  In this scenario some form of exercise would be daily where work routines could be varied, an ‘exercise regime to suit individual horse’ with ‘some form of “cross training”, (conditioning exercise not specific to racehorse training) for example basic flat work (elementary schooling exercise)

I have removed grid work

784 best of his ability 

Line 666. healthy horse has the potential to perform to the best of his or her ability yet health was negatively aligned to welfare

Tables Don’t understand  how MWS can be zero  and Best life 7, but 3 are common to both.

I have changed common to both to standard procedures. I have removed the tables and they will go into supplementary materials.

Round 2

Reviewer 1 Report

I think the manuscript has improved since the first version, it is much easier to follow. I was not able to access the supplementary material. I just have some small suggestions listed below.

L56-59. Please revise the structure of this sentence

L284. Replace thought the with thought that

L285. Add a comma after However

L312. Replace Horses needed to out with Horses need to be turned out

L344. Please use italics for all the ad-lib

L444. Replace eighth with eight

L455-456. This las sentence seems to be incomplete, please revise.

L587-L588. Wind operations was already defied previously in line 292, no need to define again here.

L615-617. Is there any reference that could support the idea that individual paddocks that allow socialization with next door horse could be appropriate or can have a positive welfare effecr.

L718. Add a comma after Nevertheless

Author Response

  Dear Reviewer 2,

Thank you for the suggestions.  I have addressed them as follows,

L56-59.  I have revised the sentence, shown in manuscript in blue text.

L284 - now L265, added thought that

L285- now L265, comma added after however

L312- Horses needed to be replaced with Horses need to be turned out

L344- ad-lib is now ad-lib

L444- now 416,replaced with eight

L455-456- now 443 Line completed with 'an'

L587-L588, now L554. Definition of wind operation removed

L615-617.  Reference added, Henderson, A.J.Z. Don't Fence Me In: Psychological Well Being for Performance Horses. Jnl. Applied. Welfare. Science 2007, 10, 309-329

L718-now 674, comma added